# Abscisic Acid and Cytokinins Are Not Involved in the Regulation of Stomatal Conductance of Scots Pine Saplings during Post-Drought Recovery

**DOI:** 10.3390/biom13030523

**Published:** 2023-03-13

**Authors:** Ilya E. Zlobin, Radomira Vankova, Petre I. Dobrev, Alena Gaudinova, Alexander V. Kartashov, Yury V. Ivanov, Alexandra I. Ivanova, Vladimir V. Kuznetsov

**Affiliations:** 1K.A. Timiryazev Institute of Plant Physiology, Russian Academy of Sciences, Botanicheskaya Street 35, 127276 Moscow, Russia; ilya.zlobin.90@mail.ru (I.E.Z.); botanius@yandex.ru (A.V.K.); ivanovinfo@mail.ru (Y.V.I.);; 2Laboratory of Hormonal Regulations in Plants, Institute of Experimental Botany, The Czech Academy of Sciences, Rozvojová 263, 16502 Prague, Czech Republic

**Keywords:** *Pinus sylvestris*, water deficit, plant hormones, water balance, 6-benzylaminopurine

## Abstract

Delayed or incomplete recovery of gas exchange after water stress relief limits assimilation in the post-drought period and can thus negatively affect the processes of post-drought recovery. Abscisic acid (ABA) accumulation and antagonistic action between ABA and cytokinins (CKs) play an important role in regulation of stomatal conductance under water deficit. Specifically, in pine species, sustained ABA accumulation is thought to be the main cause of delayed post-drought gas exchange recovery, although the role of CKs is not yet known. Therefore, we aimed to study the effects of ABA and CKs on recovery of stomatal conductance in greenhouse-grown 3-year-old Scots pine saplings recovering from water stress. We analysed both changes in endogenous ABA and CK contents and the effects of treatment with exogenous CK on stomatal conductance. Drought stress suppressed stomatal conductance, and post-drought stomatal conductance remained suppressed for 2 weeks after plant rewatering. ABA accumulated during water stress, but ABA levels decreased rapidly after rewatering. Additionally, *trans*-zeatin/ABA and isopentenyladenine/ABA ratios, which were decreased in water-stressed plants, recovered rapidly in rewatered plants. Spraying plants with 6-benzylaminopurine (0.1–100 µM) did not influence recovery of either stomatal conductance or needle water status. It can be concluded that the delayed recovery of stomatal conductance in Scots pine needles was not due to sustained ABA accumulation or a sustained decrease in the CK/ABA ratio, and CK supplementation was unable to overcome this delayed recovery.

## 1. Introduction

Drought is the most important abiotic factor threatening productivity and stability of terrestrial biomes [1]. The specific feature of trees as long-living perennial plants is their exposure to multiple, and often consecutive, drought events during their lifetime. This exposure, in turn, increases the importance of recovery processes in tree adaptation to increasing climate aridity, and scientific interest in tree post-drought recovery has been increasing in recent years [2]. Gymnosperms are characterized by higher vulnerability to subsequent droughts and more evident lagged drought effects (“legacy effects” [3]) compared to angiosperms [4,5,6,7] (an alternative point of view [8,9]), and ability to recover from drought can be more important for gymnosperms [8]. Therefore, studying factors determining the recovery processes in gymnosperms is of great interest and importance.

Among different physiological processes, gas exchange is among the most sensitive to water stress and decreases strongly even under moderate water deficit [10]. At the same time, recovery of stomatal conductance is often delayed compared to other physiological traits [2]. Delayed or incomplete recovery of gas exchange after water stress relief is often observed in various conifers [11,12,13,14]. High drought sensitivity of stomatal conductance and delayed and/or insufficient recovery of stomatal conductance together cause plants to experience a prolonged period of diminished gas exchange and assimilation, thus deteriorating plant carbon balance. Insufficiency of assimilates is unlikely to directly cause plant death during drought, but it would make plants more vulnerable to the adverse effects of the upcoming winter [15] and compromise plant growth and drought tolerance in the next growing season, which is especially important for conifers given their small fraction of stem parenchyma tissue and low stem carbohydrate storage [8]. Together, these factors make rate and completeness of gas exchange recovery the most important factor determining plant productivity and mortality in the expected future conditions with increasing frequency and severity of adverse climatic events.

Recovery of stomatal conductance after drought stress depends on several factors. In the simplest case, stomatal conductance is merely a function of leaf water status, and, therefore, recovery of stomatal conductance occurs according to the hydraulic limitation model [16]. This simple model of stomatal regulation is applicable to some more recently evolved gymnosperm species belonging to the Cupressaceae and Taxaceae clades, which rely on very low leaf water potential to keep stomata closed during prolonged water stress [17]. However, the majority of conifers, including the most widespread and important Pinaceae, rely on massive ABA accumulation to diminish stomatal conductance during water stress [17]. High concentrations of ABA accumulate during water stress, and a gradual decrease in ABA concentrations after rewatering hinders stomatal reopening [18,19], thus making ABA the clearest candidate to regulate rate and completeness of stomatal recovery after relief from water stress. However, other plant hormones can influence stomatal conductance, and cytokinins (CKs) are arguably the most conspicuous among them. In some studies, CKs were found to be ABA antagonists, suppressing ABA-induced stomatal closure and causing reopening of closed stomata [19,20], and the ratio between CKs and ABA was proposed to influence stomatal behaviour [21,22,23]. In other cases, CKs can increase stomatal conductance independently of ABA action or have no effects on stomatal conductance (Farber et al. [24] and references therein). Importantly, CK metabolism is not buffered to such an extent as ABA metabolism, and, therefore, direct manipulations of CK levels can possibly alter drought tolerance, unlike ABA accumulation, which is difficult to change [21]. CKs and their interaction with ABA can potentially influence stomatal behaviour of conifers, including during stress recovery, but this issue has not yet been studied in conifers.

In our previous study [14], we observed that Scots pine plants demonstrated incomplete recovery of stomatal conductance after long-term water stress. Since Scots pine, similar to other pines, belongs to the R-species, which continuously accumulates ABA over a period of drought stress [17], we propose that this incomplete recovery can be due to increased ABA content and/or decreased CK/ABA ratio in the needles of recovering plants. Therefore, we aimed to study the role of ABA and CKs in regulation of stomatal reopening in Scots pine by two complementary approaches:-analysis of changes in ABA and CK contents in pine plants during water stress and recovery (Experiment 1);-analysis of the effects of exogenous CK treatment on stomatal conductance in pine plants (Experiment 2).

## 2. Materials and Methods

### 2.1. Plant Material

Containerized 3-year saplings of Scots pine (*Pinus sylvestris* L.) grown from improved seeds collected in the Semyonovskoe Division of Forestry (Nizhny Novgorod Region, Russia) were purchased from the forest nursery “Semenovskij specsemleskhoz” (Nizhny Novgorod Region, Russia). Saplings were transplanted to pots (7 L for Experiment 1 and 2.5 L for Experiment 2) filled with soil substrate (peat, sand, ground limestone, 350 mg/L N, 30 mg/L P, 400 mg/L K, pH 5.5–6.0). After an overwintering period outdoors, the plants were transferred into a cold frame with a temperature of approximately 5 °C for 2 weeks and then transferred to a warmed greenhouse under natural illumination for the experiments. Plant height was approx. 50 cm. During Experiment 1, the mean midday photosynthetically active radiation level was 250 µmol/m^−2^·s^−1^, and the mean ambient temperature was 23 °C. During Experiment 2, the mean midday photosynthetically active radiation level was 175 µmol/m^−2^·s^−1^, and the mean ambient temperature was 24 °C.

### 2.2. Experimental Design

#### 2.2.1. Experiment 1

The design of Experiment 1 was described in Zlobin et al. [14]. Briefly, 3-year-old pine plants in 7 L round-shaped pots (22 cm × 25.5 cm) were divided into two groups: the first was watered as usual (“control” variant, Figure 1A), and the second was not watered (“drought” variant, Figure 1A). After 109 days of water deprivation, the latter group developed substantial water stress, which was manifested in an almost-two-fold sharp decline in needle water potential and a five-fold decrease in stomatal conductance to water vapour (g_sw_) compared with those of the control plants. This day was designated Day 0. At Day 0, the g_sw_ and needle water status, midday leaf water potential (Ψ_leaf_) and needle relative water content (RWC) were determined in control and drought-stressed plants, after which half of the drought-stressed plants were rewatered to full soil capacity (“recovery” variant, Figure 1A), whereas the remaining half faced continuous drought. At Day 0, Day 3 and Day 14 after rewatering, the g_sw_ and needle water status (Ψ_leaf_, RWC) were measured in control, drought-stressed and recovering plants in 6 biological replicates (1 plant per replicate). For more details, see Zlobin et al. [14]. Additionally, 6 biological replicates of needle samples from each experimental variant were collected, frozen in liquid nitrogen and stored at −70 °C for analysis of hormone contents.

#### 2.2.2. Experiment 2

To investigate whether treatment with CKs can influence stomatal conductance and needle water status, Experiment 2 comparing the recovery of plants with no CK spraying to plants sprayed with increasing 6-BAP concentrations (from 0.1 to 100 µM) was performed. A wide range of CK concentrations were used since CK’s effects on stomatal conductance can differ depending on CK concentration [25]. To allow plants to recover more fully than during Experiment 1, the recovery period was increased to 21 days compared to 14 days in Experiment 1.

Similar to Experiment 1, 3-year-old pine plants in 2.5 L square pots (15 cm × 15 cm × 15 cm) were divided into two groups: the first group was watered as usual (“control” variant, Figure 1B), and the second group was not watered. After 35 days of water deprivation, g_sw_ and needle water status (Ψ_leaf_, RWC) were determined in the control and drought-stressed plants. This day was designated Day 0. Next, drought-stressed plants were divided into 6 experimental groups (Figure 1B):“Drought”—plants under continuous drought, which were not rewatered at Day 0 and thereafter.“Recovery”—plants rewatered to full soil capacity and sprayed with water (12.5 mL per plant) supplemented with 0.02% Silwet™ 408 as the surfactant.“Recovery + 0.1 BAP”—plants rewatered to full soil capacity and sprayed with 0.1 µM 6-benzylaminopurine (6-BAP) in water (12.5 mL per plant) + 0.02% Silwet™ 408.“Recovery + 1.0 BAP”—plants rewatered to full soil capacity and sprayed with 1.0 µM BAP in water (12.5 mL per plant) + 0.02% Silwet™ 408.“Recovery + 10 BAP”—plants rewatered to full soil capacity and sprayed with 10 µM BAP (12.5 mL per plant) + 0.02% Silwet™ 408.“Recovery + 100 BAP”—plants rewatered to full soil capacity and sprayed with 100 µM BAP (12.5 mL per plant) +0.02% Silwet™ 408.

Each experimental group consisted of 4 biological replicates, with 3–4 plants in each replicate. The experimental duration was 21 days. g_sw_ and needle water status (Ψ_leaf_, RWC) were measured at midday on Day 3, Day 7, Day 14 and Day 21. Spraying with a handheld pulveriser and rewatering were performed in the evening on Day 0, Day 3, Day 7 and Day 14. The plants under continuous drought (“drought”, Figure 1B) died during the 3rd week of the experiment, as judged by fully desiccated foliage.

### 2.3. Stomatal Conductance to Water Vapour

Midday g_sw_ was measured using an LI-600 flow-through differential porometer (LI-COR, Inc., Lincoln, NE, USA) between 11:00 and 13:00 h. The surface area of the needles used for g_sw_ measurement was calculated, and stomatal conductance was calculated per unit of needle area. For details, see Zlobin et al. [14].

### 2.4. Leaf Water Potential (Ψ_leaf_)

Midday Ψ_leaf_ was measured using a pressure chamber instrument (Model 615, PMS Instrument Co., Albany, OR, USA). The measurements were completed between 11:00 and 13:00 h. Note that the pressure chamber instrument was able to measure the water potential down to −4.0 MPa, and, therefore, lower Ψ_leaf_ values were unmeasurable.

### 2.5. Absolute and Relative Needle Water Content

Needle RWC was determined within several hours after needle detachment and storage on ice. The needles were trimmed (approximately 1 mm) on their basal ends, weighed on an analytical balance (AB54-S, Mettler Toledo, Greifensee, Switzerland) with an accuracy of 0.1 mg and added to Eppendorf tubes with deionized water. The tubes were placed in a tightly sealed chamber with 100% relative humidity at 4 °C in darkness for 24 h. After 24 h of saturation, the needles were individually blotted briefly with filter paper, and their water-saturated mass was determined. The needles were subsequently dried at 70 °C for three days, after which their dry weight was determined. The RWC was calculated as described previously [26].

### 2.6. Analysis of Plant Hormones and Related Compounds

Frozen needle samples (approximately 10 mg FW) were mixed with cold (−20 °C) extraction solvent (1 M formic acid). The following isotope-labelled internal standards (10 pmol/sample) were then added: ^13^C_6_-IAA, ^2^H_4_-OxIAA, ^2^H_4_-OxIAA-GE (Cambridge Isotope Laboratories), ^2^H_4_-SA, ^2^H_2_-GA_19_ (Sigma–Aldrich, St. Louis, MO, USA), ^2^H_3_-PA, ^2^H_3_-DPA (NRC-PBI), ^2^H_6_-ABA, ^2^H_5_-JA, ^2^H_5_-*trans*-Z, ^2^H_5_-*trans*-ZR, ^2^H_5_-*trans*-ZRMP, ^2^H_5_-*trans*-Z7G, ^2^H_5_-*trans*-Z9G, ^2^H_5_-*trans*-ZOG, ^2^H_5_-*trans*-ZROG, ^15^N_4_-cZ, ^2^H_3_-dihydrozeatin (DZ), ^2^H_3_-DZR, ^2^H_3_-DZ9G, ^2^H_3_-DZRMP, ^2^H_7_-DZOG, ^2^H_6_-iP, ^2^H_6_-iPR, ^2^H_6_-iP7G, ^2^H_6_-iP9G and ^2^H_6_-iPRMP (Olchemim). The samples were homogenized with zirconium balls in a FastPrep-24 5G homogenizer (MP Biomedicals, Santa Ana, CA, USA) at 6 m/s for 40 s and then centrifuged for 10 min at 4 °C and 30,000× *g*. SPE Oasis HLB 96-well plates (10 mg/well, Waters) were activated with 100 µL 100% acetonitrile, followed by 100 µL H_2_O and 100 µL 1 M HCOOH. The supernatant was applied to SPE wells. The pellet was re-extracted with 100 µL 1 M HCOOH, mixed and centrifuged and the supernatant was applied to the same SPE well. The wells were washed with 100 µL of H_2_O, and samples were eluted from the wells with 50 µL of 50% acetonitrile twice using a Pressure+ 96 manifold (Biotage). Phytohormones were separated using a Kinetex EVO C18 column (2.6 µm, 150 mm × 2.1 mm, Phenomenex). The mobile phases consisted of A—5 mM ammonium acetate and 2 µM medronic acid in water and B—95:5 acetonitrile:water (*v*/*v*). The following gradient program was applied: 5% B at 0 min, 7% B at 0.1 min to 5 min, 10% to 35% B at 5.1 min to 12 min, 100% B at 13 min to 14 min and 5% B at 14.1 min. Hormone analysis was performed using an LC/MS system consisting of a UHPLC 1290 Infinity II (Agilent, Santa Clara, CA, USA) coupled to a 6495 Triple Quadrupole Mass Spectrometer (Agilent). MS analysis was performed in MRM mode using the isotope dilution method. Data processing was performed with Mass Hunter software B.08 (Agilent).

### 2.7. Statistical Analysis

During Experiment 1, six biological replicates were analysed, with 1 plant per replicate. During Experiment 2, four biological replicates were analysed, with 3–4 plants per replicate. The data were statistically analysed using SigmaPlot 12.3 (Systat Software). Pairwise comparisons of the means were performed using Student’s *t*-test for normally distributed data or Mann–Whitney rank sum test when the *t*-test was not applicable (*p* < 0.05). To calculate the strength and significance of correlations between the contents of different hormone-related compounds, Pearson’s correlation was used. The Pearson’s correlation coefficients presented in the text are significant at *p* < 0.05. The values presented in the figures are the arithmetic means, and the values presented in the tables are arithmetic means ± standard errors.

## 3. Results

### 3.1. Changes in Needle Water Status in Water-Stressed and Recovering Plants (Experiment 1)

The influence of drought stress and rewatering on pine g_sw_ and Ψ_leaf_ was described previously [14]. Briefly, drought stress resulted in substantial deterioration of water status in drought-stressed plants without rewatering (“drought”, Figure 2), with stomatal conductance decreasing from 0.023 at Day 0 to 0.003 at Day 14, leaf midday water potential decreasing from −1.62 to −2.92 MPa and RWC decreasing from 77.6 to 70.5% from Day 0 to Day 14. Following rewatering (“recovery”, Figure 2), g_sw_ did not recover to the control level. On the third and fourteenth days of recovery, stomatal conductance was 37 and 38.7% of the control, respectively. At the same time, the needle water status recovered rapidly and did not differ from that of the control by Day 14 (Figure 2; see also [14]).

### 3.2. Changes in Hormonal Balance in Water-Stressed and Recovering Plants (Experiment 1)

The increasing intensity of water stress was clearly reflected by the dynamics of ABA concentration in the needles of water-stressed plants. At Day 0, the needle ABA concentration was 2.5 times higher in “drought” than in “control” plants but increased 2.9-fold by Day 14 and exceeded the control level by 5.9-fold (Figure 3, Appendix A). In contrast, ABA content decreased in the needles of rewatered plants (“recovery”), becoming similar to the control level by Day 14 (Figure 3, Appendix A). Quite similar dynamics were observed for 7OH-ABA (*r* = 0.89 between ABA and 7OH-ABA) and to a lesser extent for ABA-Me (*r* = 0.66 between ABA and ABA-Me) (Appendix A). Notably, the concentrations of 7OH-ABA in water-stressed plants were several-fold higher than those of ABA. 9OH-ABA, PA and DPA were present in lower concentrations than ABA, responded weakly to changes in the watering regime and correlated weakly or moderately with ABA (*r* = 0.54, 0.53 and 0.38, respectively). Notably, a very high positive correlation (*r* = 0.95) was observed between 9OH-ABA and PA (Appendix A). Further, neoPA was not detected in the samples. ABA-GE was by far the most abundant ABA-related compound, being present in the range of 84–186 nmol/g DW. No clear reaction to the watering regime was present for this compound, and no significant correlation with free ABA content was observed.

The contents of *t*Z and DZ were consistently higher in drought-stressed than in control plants, especially increasing by the end of the experiment (Day 14), and tended to be higher in plants recovering from water deficit (Figure 3, Appendix A). The *t*Z content correlated with *t*ZR (*r* = 0.45), *t*ZOG (*r* = 0.52) and *t*ZROG (*r* = 0.34), whereas the DZ content correlated with DZRMP (*r* = 0.30), DZR (*r* = 0.50) and DZOG (*r* = 0.44) (Appendix A). In contrast to *t*Z and DZ, the iP content was more stable and was not consistently influenced by drought stress or recovery. The iP content did not correlate significantly with either the iPRMP or iPR content (Appendix A). Free *c*Z was not found in the pine needles; *c*Z precursors and metabolites (*c*ZRMP, *c*ZR, *c*ZOG, *c*ZROG) did not depend substantially on water supply, except for *c*ZRMP, which was higher both in drought-stressed and in recovering plants at Day 0 and decreased gradually by Day 14 (Figure 3, Appendix A).

The *t*Z/ABA ratio tended to be higher in the control than in the drought-stressed plants at Day 0, but the high variation in the ratio between the control plants made the difference nonsignificant (Figure 3, Appendix A). However, at Day 3 and Day 14, the *t*Z/ABA ratio was significantly lower in drought-stressed plants than in both control and rewatered plants, with the latter two having similar *t*Z/ABA ratios. Similar but more pronounced dynamics were observed for the iP/ABA ratio. In contrast, the DZ/ABA ratio did not demonstrate consistent changes in response to water stress and recovery, except for an almost three-fold increase in rewatered plants at Day 3.

### 3.3. Effects of Exogenous Cytokinin on Recovery of Needle Water Status (Experiment 2)

By Day 0, the needle Ψ_leaf_ in drought-stressed plants was equal to −2.11 MPa (Figure 4A). Similar values were observed at Day 3 and Day 7, but, at Day 14, a very large drop in Ψ_md_ down to unmeasurable values (<−4.0 MPa) was observed, and, therefore, data are absent for this time point (see Section 2). In the following third week of the experiment, all plants under continuous drought died, and, therefore, data for Day 21 are also absent. In all recovering plants, a consistent increase in Ψ_leaf_ was observed from Day 3 to Day 14, and, by Day 21, the needle Ψ_leaf_ was within the range between −1.1 and −1.3 MPa, compared to −0.88 in control plants (Figure 4A). Further, 6-BAP spraying did not influence the Ψ_leaf_ recovery dynamics between rewatered variants (Figure 4A).

By Day 0, the drought-stressed plants had decreased RWC (77.2%) compared to the control plants (83.6%) (Figure 4B). During the subsequent 2 weeks, the RWC of drought-stressed plants decreased promptly and consistently and reached 49.0% by Day 14. In the rewatered plants, recovery of leaf RWC was evident, and no significant differences between control plants and all recovering variants were present at Day 14 and Day 21. As with Ψ_leaf_, no consistent influence of 6-BAP spraying on RWC recovery was observed (Figure 4B).

Midday g_sw_ dropped to 8.7% of the control level by Day 0 (Figure 4C). In the plants under continuous drought, it remained strongly suppressed until plant death during the third week. In all recovering variants, g_sw_ remained lower than that of the control during the first two weeks of recovery, similar to Experiment 1, but increased from Day 14 to Day 21, and no significant differences were observed between the control and recovery variants at Day 21. Further, 6-BAP spraying did not influence g_sw_ recovery in the pine plants (Figure 4C).

## 4. Discussion

### 4.1. ABA Accumulates during Water Stress and Decreases Rapidly in Recovering Plants

Stomatal conductance provides a sensitive plant-based measure of water stress severity and is used frequently to define the point at which rewatering is required [12,27]. During Experiment 1, the g_sw_ dynamics indicated that plants experienced substantial water deficit by the beginning of the experimental period (Day 0), which was exacerbated by the end of the period (Day 14). The 2-week experimental period was characterized by rapid ABA accumulation during progressive water stress, peaking by Day 14, whereas a rapid decrease in ABA accumulation was found after rewatering (Figure 2). Further, 7OH-ABA demonstrated an accumulation pattern similar to ABA (*r* = 0.89), with even more marked accumulation under progressive water stress and a higher total amount compared to ABA. This finding raises the question of the biological importance of 7OH-ABA accumulation; 7OH-ABA is considered a minor product of ABA hydroxylation [28], but, in conifers, it can be present in substantial amounts [29]. Neither the enzyme responsible for ABA C7′-hydroxylation nor further metabolism of 7OH-ABA are currently known [28,29,30]. There are different data on the biological activity of 7OH-ABA, favouring both the presence [31] and absence [32] of substantial ABA-like biological activity. However, based on the quantitative abundance of 7OH-ABA and the similarity of accumulation patterns between 7OH-ABA and ABA, we propose that 7OH-ABA may have a substantial biological role in pine reaction to water deficit. The third ABA metabolite demonstrating a water-stress-dependent accumulation pattern is ABA-methyl ester, but its levels were at least an order of magnitude lower than those of ABA or 7OH-ABA. Importantly, the decrease in accumulation of ABA, 7OH-ABA and ABA-Me in recovering plants argued against the idea that sustained ABA accumulation underlies impaired recovery of stomatal conductance.

Other oxidative ABA metabolites (9OH-ABA, PA, DPA) were present in substantially lower amounts compared to ABA and 7OH-ABA and demonstrated no clear reaction to water stress and recovery. The observed very tight correlation between 9OH-ABA and PA contents (*r* = 0.95) was unexpected since neoPA and not PA is the metabolite of 9OH-ABA. However, neoPA was not found in pine plants. Considering ABA-GE, this compound was by far the most abundant ABA metabolite in both the experimental plants and naturally grown forest pines [33]. Most likely, ABA-GE accumulation increases with plant maturity since this compound was present in minor amounts in the juvenile needles of 8-week-old Scots pine seedlings [34].

### 4.2. Dynamics of Cytokinins and CK/ABA Ratio Do Not Coincide with Recovery of Stomatal Conductance 

Among three free active CKs found in pine needles, *t*Z and iP were shown to bind with high affinity to *Picea abies* CHK receptors, whereas PaCHK affinity for DZ was one to two orders of magnitude lower [35]. Therefore, the dynamics of *t*Z and iP are likely to have more biological significance than DZ. The *t*Z content was higher in plants under continuous drought and to a lesser extent in recovering plants, whereas no consistent influence of water supply on iP dynamics was observed. However, both of these CKs, unlike DZ, demonstrated a rather similar link with ABA dynamics since both the *t*Z/ABA and iP/ABA ratios were lowered in water-stressed plants but recovered promptly after rewatering. This increase in *t*Z/ABA and iP/ABA ratios was not followed by increasing plant stomatal conductance, which remained 60% lower than the control in Experiment 1. Moreover, during Experiment 2, spraying with different concentrations of 6-BAP influenced neither recovery of needle water status (Ψ_leaf_, RWC) nor recovery of stomatal conductance, and these parameters recovered by the end of the 3-week recovery period independently of CK spraying. It is important to note that 6-BAP is bound by CHK receptors of *Picea abies* plants with an affinity intermediate between those of *t*Z/iP and DZ [35], and it is reasonable to expect that 6-BAP can also bind to CHK receptors in pine needles. It can be concluded that partial suppression of stomatal opening in pine plants is unresponsive to decreased ABA accumulation, increased CK content and CK/ABA ratio and exogenous CK treatment, which implies that this suppression is not mediated by sustained effects of water stress on plant hormonal balance.

### 4.3. Possible Reasons for Delayed Recovery of Stomatal Conductance

If not hormone-mediated, how can suppression of g_sw_ recovery be explained? One potential explanation is the decreased hydraulic supply due to decreased leaf hydraulic conductance [16]. Water-stress-induced changes in leaf hydraulic conductance can be due to various reasons, including leaf xylem cavitation, reversible xylem collapse, shrinkage of mesophyll cells and changes in membrane water permeability [2,36,37]. The decrease in outside-xylem hydraulic conductance is often responsible for the majority of the decline in leaf hydraulic conductance during dehydration [37,38], but these extraxylary declines in conductance are generally rapidly recoverable and, therefore, cannot be responsible for the sustained depression of stomatal conductance for several weeks after drought relief [39]. In contrast, leaf xylem embolism is likely a critical trait for predicting the legacy of drought in plant communities [40]. We did not measure leaf xylem embolism directly, but we can arguably propose that substantial embolism was unlikely to be present during either experiment. Sustained leaf xylem embolism leads to hydraulic failure and irreversible desiccation in downstream mesophyll tissues, which lose the ability to rehydrate [41,42] and can therefore be assessed by loss of leaf rehydration capacity. Recent studies have shown that a 10% loss of leaf rehydration capacity occurs when leaf RWC drops to lower than 70%, although differences between species are present [43,44]. In both Experiment 1 and Experiment 2, the needle RWC was higher than 77% at Day 0 before rewatering and increased further after rewatering, achieving the control level by Day 14. In addition, leaf water potential was quite high in drought-stressed pine plants at Day 0 before rewatering in both Experiment 1 (−1.62 MPa) and Experiment 2 (−2.11 MPa). Moreover, the latter two studies [43,44] examined only angiosperm species, and gymnosperms have generally less vulnerable leaf hydraulic conductance than angiosperms [37]. Therefore, the degree of needle dehydration experienced by rewatered plants in Experiment 1 and Experiment 2 is not expected to be enough to induce substantial leaf xylem embolism and, therefore, the long-lasting depression of stomatal conductance after stress relief. Depression of stomatal conductance can probably be induced by persistent stress effects on transcription of genes involved in stomatal regulation, which remains even after stomatal ABA concentration retrieval to control values [45]. The reasons for sustained depression of stomatal conductance in recovering conifers clearly deserve further study.

## 5. Conclusions

Both drought stress and recovery exerted substantial effects on the balance of CKs and ABA-related compounds in Scots pine plants. ABA and 7OH-ABA demonstrated similar dynamics, with prominent increases under water stress and decreases after stress relief. The *t*Z/ABA and iP/ABA ratios decreased under water stress and increased after rewatering, thus suggesting opposing roles of ABA and CKs in adaptation to drought stress and in post-stress recovery. However, changes in the CK/ABA ratio were unrelated to the dynamics of recovery of stomatal conductance of Scots pine plants, and exogenous CK treatment influenced neither recovery of stomatal conductance nor needle water status. Therefore, sustained depression of stomatal conductance after water stress relief cannot be explained by sustained ABA accumulation or by a shift in the CK/ABA ratio. To date, the pattern of ABA accumulation is thought to play a pivotal role in regulation of dynamics of stomatal conductance under water stress [17]. Our findings question the significance of changes in hormonal balance in restriction of stomatal conductance during post-drought recovery. More in-depth study of causes of hampered recovery is required to contribute to mechanistic models of tree performance under water stress. 

## Figures and Tables

**Figure 1 biomolecules-13-00523-f001:**
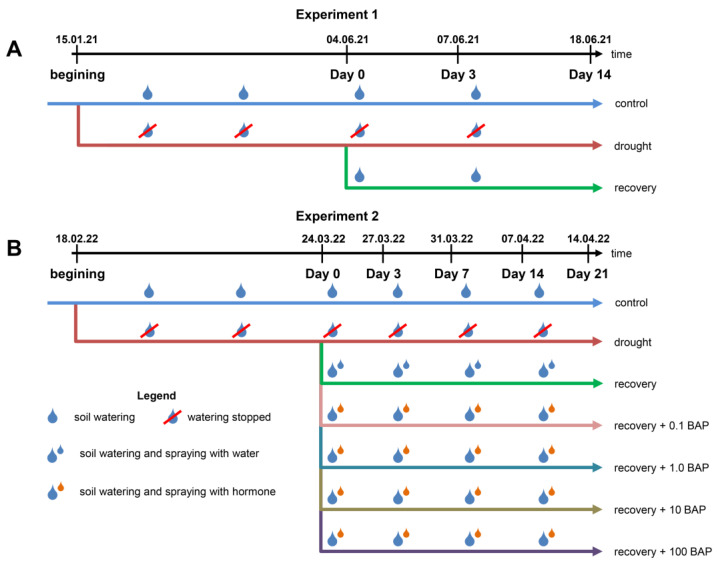
The design of Experiment 1 (**A**) and Experiment 2 (**B**). The scheme design of Experiment 1 was adapted from Zlobin et al. [14]. BAP—6-benzylaminopurine.

**Figure 2 biomolecules-13-00523-f002:**
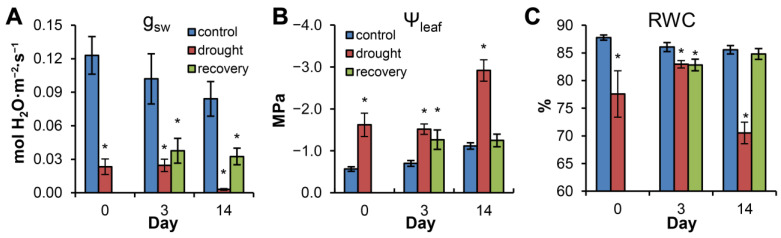
Effects of water stress on stomatal conductance (**A**), leaf water potential (**B**) and RWC (**C**) in the needles of plants in Experiment 1. The experimental data are adapted from Zlobin et al. [14]. Pairwise comparisons of the means were performed between control and drought-stressed and control and recovering plants at each separate time point using Student’s *t-*test or Mann–Whitney rank sum test. Asterisks (*) denote significant differences at *p* < 0.05 between the control and experimental variants at each time point.

**Figure 3 biomolecules-13-00523-f003:**
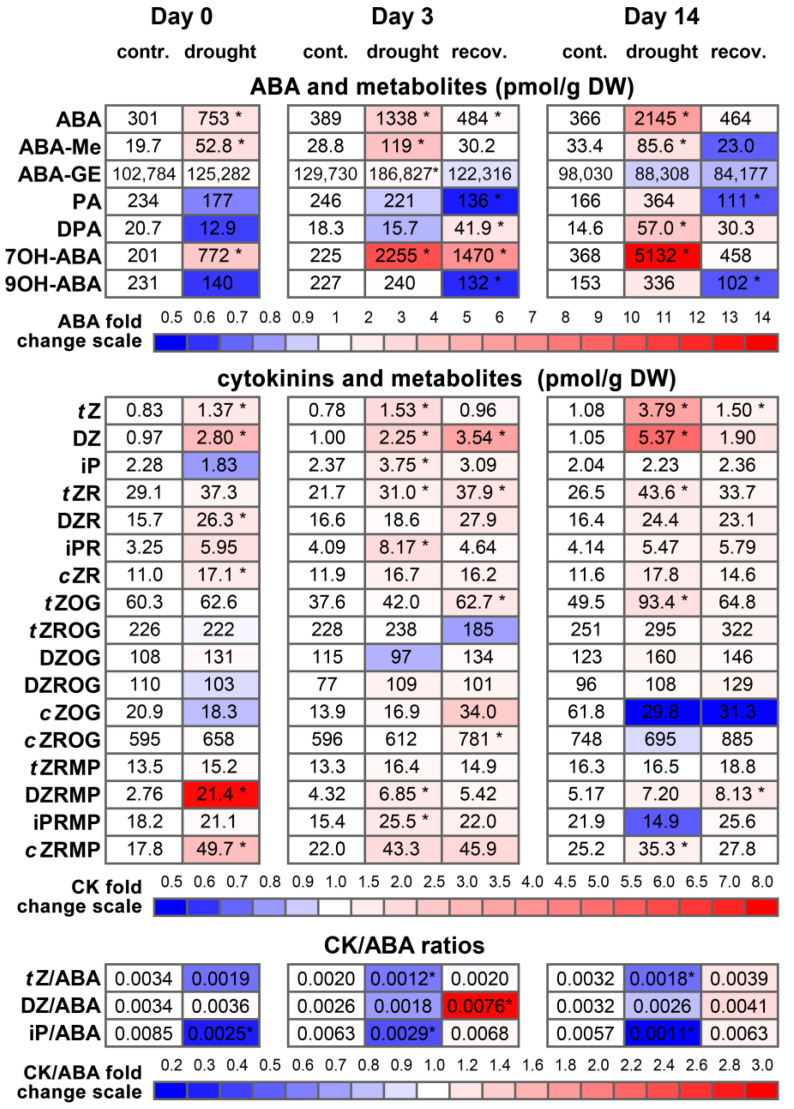
Heatmap analysis of the contents of abscisic acid, cytokinins and their metabolites and of CK/ABA ratios in the needles of plants in Experiment 1. The contents of the compounds in control plants at each time point were taken as 1.0 (white); the relative increase is indicated in red and the relative decrease is indicated in blue. ABA, abscisic acid; ABA-Me, ABA-methyl ester; ABA-GE, ABA-glucosyl ester; PA, phaseic acid; DPA, dihydrophaseic acid; 7OH-ABA, 7-hydroxy-ABA; 9OH-ABA, 9-hydroxy-ABA; *t*Z, *trans*-zeatin; DZ, dihydrozeatin; iP, N^6^-(Δ^2^-isopentenyl)adenine; *t*ZR, *trans*-zeatin 9-riboside; DZR, dihydrozeatin 9-riboside; iPR, N^6^-(∆^2^-isopentenyl)adenosine; *c*ZR, *cis*-zeatin 9-riboside; *t*ZOG, *trans*-zeatin O-glucoside; *t*ZROG, *trans*-zeatin 9-riboside O-glucoside; DZOG, dihydrozeatin O-glucoside; DZROG, dihydrozeatin 9-riboside O-glucoside; *c*ZOG, *cis*-zeatin O-glucoside; *c*ZROG, *cis*-zeatin 9-riboside O-glucoside; *t*ZRMP, *trans*-zeatin 9-riboside-5′-monophosphate; DZRMP, dihydrozeatin 9-riboside-5′-monophosphate; iPRMP, N^6^-(∆^2^-isopentenyl)adenosine-5′-monophosphate; *c*ZRMP, *cis*-zeatin 9-riboside-5′-monophosphate. Pairwise comparisons of the means were performed between control and drought-stressed and control and recovering plants at each separate time point using Student’s *t-*test or Mann–Whitney rank sum test. Asterisks (*) denote significant differences at *p* < 0.05 between the control and experimental variants at each time point.

**Figure 4 biomolecules-13-00523-f004:**
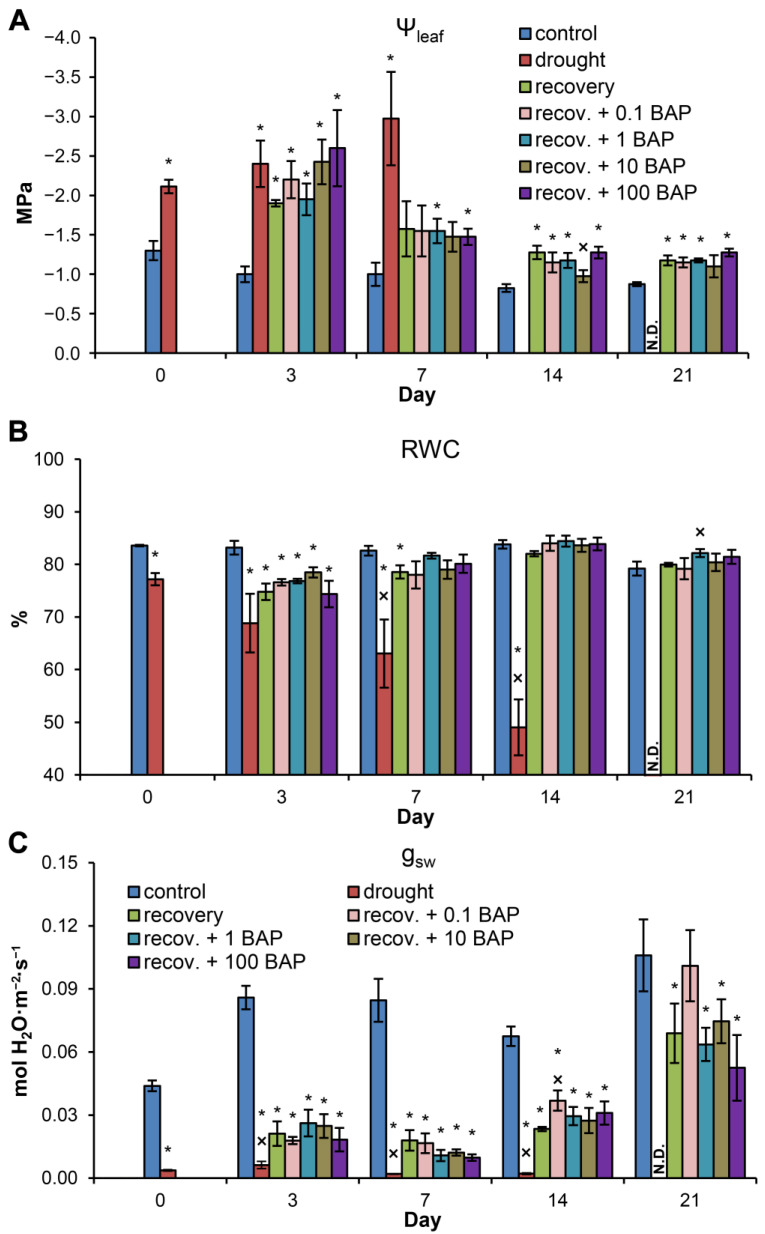
The influence of water deficit, recovery and 6-BAP spraying on the leaf water potential (**A**), RWC (**B**) and stomatal conductance (**C**) of pine plants during Experiment 2. In the first set of pairwise comparisons of the means, all variants were compared with “control” plants at each separate time point using Student’s *t-*test or Mann–Whitney rank sum test. Asterisks (*) denote significant differences at *p* < 0.05 between the control and experimental variants at each time point. In the second set of pairwise comparisons of the means, all variants were compared with “recovery” plants at each separate time point using Student’s *t-*test or Mann–Whitney rank sum test. Multiplication signs (×) denote significant differences at *p* < 0.05 between the “recovery” variant and the other experimental variants at each time point. N.D.—plants are dead, and, therefore, no data are present.

## Data Availability

The datasets generated and/or analyzed during the current study are available from the corresponding author on reasonable request.

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
