# Peer review of "Abscisic Acid and Cytokinins Are Not Involved in the Regulation of Stomatal Conductance of Scots Pine Saplings during Post-Drought Recovery"

_biomolecules, 2023, doi:10.3390/biom13030523_

Round 1

Reviewer 1 Report

Dear Authors, 

A very nice work and well presented. However there is a fundamental point which I would like to mention. The experiments are pot experiments and does not mimic the actual environment. gsw, Ψleaf, are the consequences and manifestations of drought stress, but not a direct measurement of drought. Choosing 109 days of water deprivation as a start point of drought stress based on physiological changes does not seem a correct scientific approach.

I would recommend measuring the soil water content (SWC) regularly from the beginning, and choosing drought start and other drought points based on SWC, rather than choosing as day 0, 3, 7..... This might result in drought points to be on different days for different pots/plants. However, this will give more information about drought intensity at which the parameters were measured. Moreover, the measurements will be comparable as taken at the same drought intensity. 

Results: Recommended to be paragraphed as findings with proper sub-headings, and not as Exp 1 and 2. 

Discussion: Well written, however, recommended to be paragraphed with proper sub-headings. 

Although well written and debated, mostly the discussion and conclusion is pretty much mixed. Recommended to have a separate conclusion with heading.  

Author Response

Dear Authors, 

Comment. A very nice work and well presented. However there is a fundamental point which I would like to mention. The experiments are pot experiments and does not mimic the actual environment. gsw, Ψleaf, are the consequences and manifestations of drought stress, but not a direct measurement of drought. Choosing 109 days of water deprivation as a start point of drought stress based on physiological changes does not seem a correct scientific approach.

Answer: Dear Reviewer, we appreciate the attentive review and evaluation of our manuscript. We considered all of your comments and suggestions and improved our manuscript.

Comment. I would recommend measuring the soil water content (SWC) regularly from the beginning, and choosing drought start and other drought points based on SWC, rather than choosing as day 0, 3, 7..... This might result in drought points to be on different days for different pots/plants. However, this will give more information about drought intensity at which the parameters were measured. Moreover, the measurements will be comparable as taken at the same drought intensity. 

Answer: In our study, we utilized the classical definition of “stress” introduced by Hans Selye, who described stress as an organism’s response to an adverse environmental factor. We argue that plant-based parameters are the direct characteristic of plant physiological state under drought, since the water stress by itself is the characteristic of the plant and not of the soil on which plant is grown. The actual availability of water in the soil would depend largely on soil characteristics – for example, water availability for the plant would be higher on sandy soils compared to loam soils at the given soil volumetric water content. Therefore, we could not use changes in soil volumetric content to define the point at which plants experienced an adequate level of water stress. Moreover, during destructive plant harvesting, we observed that soil was frequently wet in the basal part of the pots, but the roots of pine plants were unable to reach this water, and plants were severely stressed. In contrast, the decreases in stomatal conductance and RWC are reliable plant-based indicators of water stress. The rapid drop in stomatal conductance, which occurred after 109 days of water deprivation, clearly indicated the increased intensity of water stress. The same approach is often used. For example, Rehschuh et al. (Plant Physiology, 2020) continued water stress until stomatal conductance dropped to almost zero, and Fox et al. (Tree Physiology, 2018) continued stress until stomatal conductance was sharply reduced before performing rewatering. In our earlier study on field-grown plants (Pashkovskiy et al., Environmental and Experimental Botany, 2022), we relied on soil volumetric content relative to field capacity (in %) to indirectly estimate the extent of water stress experienced by pine and spruce trees, but this was motivated by our inability to directly measure the stomatal conductance in the field-grown trees due to spatial restrictions.

Comment. Results: Recommended to be paragraphed as findings with proper sub-headings, and not as Exp 1 and 2. 

Answer: According to your recommendations, we extended the subheadings of each paragraph in the Results section.

Comment. Discussion: Well written, however, recommended to be paragraphed with proper sub-headings. 

Although well written and debated, mostly the discussion and conclusion is pretty much mixed. Recommended to have a separate conclusion with heading.  

Answer: According to your recommendations, we separated the Conclusion section in the improved version of the Manuscript. Considering the subheadings in the Discussion section, we examined several dozen articles published in Biomolecules and found that the subsectioning of the Discussion section is generally not practiced. Therefore, we decided not to subsection the Discussion section in our Manuscript.

Reviewer 2 Report

Biomolecules-2190076

This MS studies the effects of ABA and CKs on the regulation of stomatal conductance in P. sylvestris 3-years old plants  during post-drought recovery. The topic is interesting, but the MS needs a thorough revision in all sections: Introduction needs to be better focused, information is missing in Mat&Met, statistical analysis needs further explanation, I suggest to mark significant results and to include a correlation matrix, Discussion is interesting but needs revision, some conclusions are missing.

Some parts are not easy to follow. I suggest to make the English revised by a proffesional

Abstract

-I miss a description of the treatments, the type of plant material that is used, where is the experiment done (greenhouse).

- L19-21: please revise sentence

L21: use plants, instead of saplings

Keywords

-I suggest not to include as keywords those words that are already mentioned in the title.

Introduction

I suggest the authors to make  a thorough revision of this section, as it is not well focused.

-L37-39: revise sentence

-L39-42: I don’t really get this Gymnosperms vs Angiosperms sentences. In this study authors focus just in one species. Why is of great interest and importance?........I suggest to remove this part.

-L51: ..” thus thinning the plant carbon balance”….. what does this mean?

-L44-59: this paragraph is not easy to follow and its quite general. Probably, a rewiew of the English by a profesional will help. Nevertheless, I suggest to have a deep thought if this paragraph is necessary or can be reduced.

L56-59: add some reference

L81 and elsewhere: better use “conifer”

L84: please, be careful with your statements. In the mentioned paper only P. sylvestris is studied, not all pines.

L86: what are the R-species?

L91: please, indicate age of the plants, the type of experiment

Mat&Met

This sections needs a thorough revision. The experiments are not easy to understand, many information is missing. Please explain why two experiments are done and what dot you expect to obtain from each of them.

-I miss information about:  (i) tree cultivation, (ii) where was the experiment done (greenhouse, …), (iii) environmental conditions, (iv) duration of the experiment, (v) origin of the seeds, (vi) not even the species is mentioned, (vii) initial  height of plants when the experiment started, (viii) no. Plants per treatment

-Fig 1: the experiment is done in January. Can can this affect the results? Where were the plants grown? What were the environmental conditions (temp, humidity)

-Is there any data about soil moisture content?

-L101: plants were not irrigated for 109 days? Were there differences in tree growth between controls and drought plants after more tan 3 months?

L102: please quantify (give values) ; “substantial water stress, which was manifested in an almost twofold sharp 102

decline in needle water potential and a fivefold decrease in stomatal conductance to wa- 103

ter vapour (gsw) compared with those of control plants”

L112: “ from each variant”… what does this mean?

-I miss what is the objetive of each experiment

-L127: 0.02% Silwet™ 408 as the surfactant: please explain what is this for

- L129: μM 6-benzylaminopurine (6-BAP): please explain what is this for

-L136 and elsewher: “biological replicates”..what does this mean?

-L142 and below: when (date) were all the measurements done?

- L149: “according to the manufacturer’s instructions”…………..please delete

-L153 and L163: does this mean that plants were harvested? Please, explain when and how many plants

-L188: statistical analysis needs more explanation: which varaibles, factors, type of analysis, which correlations.

Results

-L201: “…….was described earlier”… what does this mean? Have results already been published? If this is so, I suggest to remove it from this MS.

-L201-209: please show level of significance

-Fig 3. What does the scale mean?

-L271-274: this should be explained in Mat&Met, not in Results

-L294 and elsewhere: avoid starting a sentence with and acronym

- I miss a correlation matrix tha will help to understand the Discussion

Discussion

-I miss a discussion of results shown in Fig 2

-L314: please, include results from  the correlation in the Results section

-L317-323: this seems more Introduction tan Discussion

-311-318: please revise this first paragraph. I cannot find really a discussion in it.

-L358-372: please revise, it seems more an Introduction tan discussion of your own results.

-L380: these are your results. What does the reference explain?

-L381: as stated by the authors, these references are not applicable to this study because behaviour of angiosperms is different to gymnosperms. In addition,  the objetive of the present MS is not to compare angiosperms vs gymnosperms……………… please revise, and I suggest to remove.

-I miss some conclusions. What have we learned form this study? Currently, it is not clear to me

Author Response

Comment: This MS studies the effects of ABA and CKs on the regulation of stomatal conductance in P. sylvestris 3-years old plants during post-drought recovery. The topic is interesting, but the MS needs a thorough revision in all sections: Introduction needs to be better focused, information is missing in Mat&Met, statistical analysis needs further explanation, I suggest to mark significant results and to include a correlation matrix, Discussion is interesting but needs revision, some conclusions are missing.

Some parts are not easy to follow. I suggest to make the English revised by a proffesional

Answer: Thank you for your consideration. Our manuscript has been revised according to your comments. The manuscript received language and style editing by American Journal Experts (Certificate Number 8927-6BE9-866C-D780-C6EF verification page at www.aje.com/certificate).

Comment: Abstract

-I miss a description of the treatments, the type of plant material that is used, where is the experiment done (greenhouse).

Answer: We added a description of the plant material and growth conditions in the Abstract.

Comment: - L19-21: please revise sentence

Answer: This sentence was separated into two sentences and reformulated.

L21: use plants, instead of saplings

Answer: We argue that the use of the term “sapling” is justified for 3-year-old Scots pine plants.

Comment:  Keywords

-I suggest not to include as keywords those words that are already mentioned in the title.

Answer: According to your recommendations, we changed the keywords.

Comment: Introduction

I suggest the authors to make a thorough revision of this section, as it is not well focused.

-L37-39: revise sentence

Answer: The language was improved by native English-speaking editors at American Journal Experts.

Comment: -L39-42: I don’t really get this Gymnosperms vs Angiosperms sentences. In this study authors focus just in one species. Why is of great interest and importance?........I suggest to remove this part.

Answer: This comparison is required to highlight the particular significance of recovery processes in gymnosperms compared with angiosperms, thus justifying the necessity to study the recovery processes in gymnosperm plants.

Comment: -L51: ..” thus thinning the plant carbon balance”….. what does this mean?

Answer: We changed “thinning” to “deteriorating”.

Comment: -L44-59: this paragraph is not easy to follow and its quite general. Probably, a rewiew of the English by a profesional will help. Nevertheless, I suggest to have a deep thought if this paragraph is necessary or can be reduced.

Answer: The language was improved by American Journal Experts (Certificate Number 8927-6BE9-866C-D780-C6EF verification page at www.aje.com/certificate). However, we argue that this paragraph is of great importance since it provides the following rationale: 1. Study the recovery of gas exchange among recovery of other physiological processes; and 2. Study of coniferous plants.

Comment: L56-59: add some reference

Answer: This sentence is not taken from a reference; instead, it is our own conclusion from the literature cited previously in the paragraph.

Comment: L81 and elsewhere: better use “conifer”

Answer: We changed “coniferous species” to “conifers” throughout the text.

Comment: L84: please, be careful with your statements. In the mentioned paper only P. sylvestris is studied, not all pines.

Answer: We added “Scots” to “pine”.

Comment: L86: what are the R-species?

Answer: The term “R-type species” corresponds to plants that continuously accumulate ABA over the period of drought stress. This term was implemented by Brodribb et al. (Proceedings of the National Academy of Sciences, 2014).

Comment: L91: please, indicate age of the plants, the type of experiment

Answer: The extended description of plant material was added to the Materials and Methods (Section 2.1).

Mat&Met

Comment: This sections needs a thorough revision. The experiments are not easy to understand, many information is missing. Please explain why two experiments are done and what dot you expect to obtain from each of them.

Answer: In the revised version of the Manuscript, we explained that Experiment 1 aimed to analyse changes in ABA and CK contents in Scots pine plants during water stress and recovery, whereas Experiment 2 aimed to analyse the effects of exogenous CK treatment on stomatal conductance in Scots pine plants (see the end of the Introduction section).

Comment: -I miss information about: (i) tree cultivation, (ii) where was the experiment done (greenhouse, …), (iii) environmental conditions, (iv) duration of the experiment, (v) origin of the seeds, (vi) not even the species is mentioned, (vii) initial height of plants when the experiment started, (viii) no. Plants per treatment

Answer: We added Section 2.1 describing the species name, plant material, origin of the seeds, environmental conditions and initial height of plants. The duration of the experiments is given in Figure 1; the total duration of Experiment 1 was 123 days (from 15.01.21 to 18.06.21), and the total duration of Experiment 2 was 56 days (from 18.02.22 to 14.04.22). In Experiment 1, the number of plants was 6 (each plant was treated as a biological replicate) per treatment at each time point. In Experiment 2, the number of plants was 12-16 (4 biological replicates with 3-4 plants in biological replicate) per treatment at each time point.

Comment: -Fig 1: the experiment is done in January. Can can this affect the results? Where were the plants grown? What were the environmental conditions (temp, humidity)

Answer: Since the plants overwintered at negative temperatures during a sufficient period, with a subsequent dormancy break, we propose that a shift in the growing period did not substantially affect our results. The environmental conditions are described in Section 2.1 of the revised version of the Manuscript.

Comment: -Is there any data about soil moisture content?

Answer: We assessed the water loss from pots during the experimental period and published these data in Fig. S3 in Zlobin et al. (Physiologia Plantarum, 2022).

Comment: -L101: plants were not irrigated for 109 days? Were there differences in tree growth between controls and drought plants after more tan 3 months?

Answer: Yes, plants were not irrigated for 109 days. However, due to the large amount of soil in pots (7 L), the buildup of water stress was quite gradual. Unfortunately, we never analysed plant growth parameters since this work was beyond the object of our study.

Comment: L102: please quantify (give values) ; “substantial water stress, which was manifested in an almost twofold sharp decline in needle water potential and a fivefold decrease in stomatal conductance to water vapour (gsw) compared with those of control plants”

Answer: The data for stomatal water conductance and needle water potential at Day 0 are presented in Figure 2.

Comment: L112: “ from each variant”… what does this mean?

Answer: We added the word “experimental”.

Comment: -I miss what is the objetive of each experiment

Answer: To clarify the objectives of each experiment, we added descriptions of Experiment 1 and Experiment 2 at the end of the Introduction section.

Comment: -L127: 0.02% Silwet™ 408 as the surfactant: please explain what is this for

Answer: Silwet™ 408 is a surfactant aimed at assisting the penetration of 6-benzylaminopurine through the needle cuticle.

Comment: -L129: μM 6-benzylaminopurine (6-BAP): please explain what is this for

Answer: 6-benzylaminopurine (6-BAP) was used as a model compound with well-established cytokinin activity (see, e.g., Wang et al. 2003, Plant and Soil; Ahmadi Lahijani et al. 2018, J. Agr. Sci. Tech.) to study the effects of cytokinin spraying on needle stomatal conductance during Experiment 2.

Comment: -L136 and elsewher: “biological replicates”..what does this mean?

Answer: Each biological replicate was treated as an independent measurement during statistical analysis. During Experiment 1, six biological replicates were analysed, with 1 plant per replicate. During Experiment 2, four biological replicates were analysed, with 3-4 plants per replicate.

Comment: -L142 and below: when (date) were all the measurements done?

Answer: During Experiment 1, all measurements were performed on Day 0 (4.06.21, see Figure 1), Day 3 (07.06.21) and Day 14 (18.06.21). During Experiment 2, all measurements were performed on Day 0 (24.03.22), Day 3 (27.03.22), Day 7 (31.03.22), Day 14 (07.03.22), and Day 21 (14.04.22).

Comment: -L149: “according to the manufacturer’s instructions”…………..please delete

Answer: The text was deleted.

Comment: -L153 and L163: does this mean that plants were harvested? Please, explain when and how many plants

Answer: In the description of Experiment 1 (Section 2.1.1), 6 plants were destructively harvested as independent biological replicates per treatment at each time point, and needle samples were taken from each plant.

Comment: -L188: statistical analysis needs more explanation: which varaibles, factors, type of analysis, which correlations.

Answer: We clarified that the correlation analysis was used not for all variables but only for the analysis of hormone-related compounds. The results of the correlation analysis were added as Supplementary tables 2 and 3.

Results

Comment: -L201: “…….was described earlier”… what does this mean? Have results already been published? If this is so, I suggest to remove it from this MS.

 Answer: The results of stomatal conductance and leaf water status in Experiment 1 were published in Zlobin et al. (Physiologia Plantarum, 2022), and the observed hysteresis in the recovery of stomatal conductance motivated us to perform the current study. However, in the current study, it was necessary to present the data on stomatal conductance and leaf water status in brief form to characterize the plant physiological status, with reference to Zlobin et al. (Physiologia Plantarum, 2022). We completely changed the form of data presentation by creating a new Figure 2. At the same time, we indicated that the data “were adapted from Zlobin et al. [13]” – our previously published manuscript.

Comment: -L201-209: please show level of significance

Answer: All differences were treated as statistically significant at the level of p < 0.05. The corresponding explanation is given in Subsection 2.7 in the revised version of the Manuscript.

Comment: -Fig 3. What does the scale mean?

Answer: In Figure 3, numbers indicate the absolute content of compounds. The color scale describes the ratio of the contents of a given compound between the experimental variant and control at a given time point. The contents of the compounds in control plants at each time point were taken as 1.0 (white), the relative increase is indicated by red, and the relative decrease is indicated by blue.

Comment: -L271-274: this should be explained in Mat&Met, not in Results

Answer: The corresponding text fragment was transferred from the Results section to the Materials and Methods section.

Comment: -L294 and elsewhere: avoid starting a sentence with and acronym

Answer: The text was improved.

Comment: - I miss a correlation matrix tha will help to understand the Discussion

Answer: The tables with correlation matrices were added to the Supplementary material (Tables S2 and S3).

Discussion

Comment: -I miss a discussion of results shown in Fig 2

Answer: These results were thoroughly discussed in Zlobin et al. (2022). In the current Manuscript, these results are presented only to improve the understanding of the original data of the current article presented in Figure 3.

Comment: -L314: please, include results from the correlation in the Results section

Answer: This value (0.89) was already given in the Results section; the presence of this value in the Discussion section was motivated by the significance of the observed link between 7OH-ABA and ABA for discussion.

Comment: -L317-323: this seems more Introduction tan Discussion

Answer: The discussion of possible biological roles of 7OH-ABA was motivated by the observed high concentrations of this compound in pine needles and therefore must be located in the Discussion section.

Comment: -311-318: please revise this first paragraph. I cannot find really a discussion in it.

Answer: The paragraph was reformulated in the revised version of the Manuscript.

Comment: -L358-372: please revise, it seems more an Introduction tan discussion of your own results.

Answer: This discussion was possible only after demonstration that the dynamics of ABA and cytokinins are not linked with the dynamics of stomatal conductance and needle water status. Therefore, it must be present in the Discussion section.

Comment: -L380: these are your results. What does the reference explain?

Answer: This sentence was reformulated.

Comment: -L381: as stated by the authors, these references are not applicable to this study because behaviour of angiosperms is different to gymnosperms. In addition, the objetive of the present MS is not to compare angiosperms vs gymnosperms……………… please revise, and I suggest to remove.

Answer: This text fragment was given to demonstrate that the hydraulic vulnerability of gymnosperms is lower than that of angiosperms, and therefore, irreversible xylem embolism in needles was unlikely in the studied plants.

Comment: -I miss some conclusions. What have we learned form this study? Currently, it is not clear to me

Answer: We added the Conclusion section to the revised version of the Manuscript.

Reviewer 3 Report

The manuscript by Zlobin et al proposes  an experiment to describe the the rule of abscisic acid and cytokinins  in the regulation of stomatal conductance of Post-Drought Recovery in scots pine saplings. The experiment is well organized and described. I have just few comments:

Introduction: Please add a reference to the first sentence.

L41- ''but see?''

Conclusion are missing.

Author Response

The manuscript by Zlobin et al proposes  an experiment to describe the the rule of abscisic acid and cytokinins  in the regulation of stomatal conductance of Post-Drought Recovery in scots pine saplings. The experiment is well organized and described. I have just few comments:

Comment: Introduction: Please add a reference to the first sentence.

Answer: The reference was added.

L41- ''but see?''

Answer: This sentence was reformulated in the revised version of the Manuscript.

Comment: Conclusion are missing.

Answer: We added the Conclusion section to the revised version of the Manuscript.

Reviewer 4 Report

The manuscript biomolecules-2190076 entitled “Abscisic Acid and Cytokinins are not Involved in the Regulation of Stomatal Conductance of Scots Pine Saplings during Post-Drought Recovery” investigated the role of hormones abscisic acid and cytokinin in the extended stomatal closure after rewatering in Scots Pine. Results of two experiments are reported in the manuscript: in the first one, endogenous hormones and their ratio were compared between stressed, well-watered and recovered plants after 3 and 14 days from recovery; in the second experiment, the effect of exogenous cytokinin on stomatal conductance after rewatering was evaluated at 4 different concentrations. Results suggest that the extended stomatal closure is not caused by hormones and some alternative hypotheses are discussed in the manuscript.

The manuscript is well organized, and results are clear and properly discussed. Although, results are not conclusive, they are interesting and useful in understanding the complex mechanisms of plant stomatal control. In my opinion, the manuscript can be accepted for publication in Biomolecules after minor revisions.

In particular, in material and methods should be specified the environmental conditions of both experiment. Were they performed in a greenhouse or under open-air conditions?

Other minor comments are reported in the following lines.

Lines 39-42. Not clear. Also, the “legacy effects” should be briefly described

Lines 56-59. Add reference

Figure 4 A. In figure capture should be specified that drought stressed plants at days 14 are not represented because exceeded -4MPa and that they were died at day 21

Author Response

The manuscript biomolecules-2190076 entitled “Abscisic Acid and Cytokinins are not Involved in the Regulation of Stomatal Conductance of Scots Pine Saplings during Post-Drought Recovery” investigated the role of hormones abscisic acid and cytokinin in the extended stomatal closure after rewatering in Scots Pine. Results of two experiments are reported in the manuscript: in the first one, endogenous hormones and their ratio were compared between stressed, well-watered and recovered plants after 3 and 14 days from recovery; in the second experiment, the effect of exogenous cytokinin on stomatal conductance after rewatering was evaluated at 4 different concentrations. Results suggest that the extended stomatal closure is not caused by hormones and some alternative hypotheses are discussed in the manuscript.

Comment: The manuscript is well organized, and results are clear and properly discussed. Although, results are not conclusive, they are interesting and useful in understanding the complex mechanisms of plant stomatal control. In my opinion, the manuscript can be accepted for publication in Biomolecules after minor revisions.

Answer: Thank you for your evaluation of our work and for your valuable comments. We take into account all your recommendations and improved the manuscript.

Comment: In particular, in material and methods should be specified the environmental conditions of both experiment. Were they performed in a greenhouse or under open-air conditions?

Answer: The description of experimental conditions was improved in the revised version of the Manuscript (see Section 2.1).

Other minor comments are reported in the following lines.

Comment: Lines 39-42. Not clear. Also, the “legacy effects” should be briefly described

Answer: The description of legacy effects and the reference were added.

Comment: Lines 56-59. Add reference

Answer: This sentence is not taken from a reference; instead, it is our own conclusion from the literature cited previously in the paragraph.

Comment: Figure 4 A. In figure capture should be specified that drought stressed plants at days 14 are not represented because exceeded -4MPa and that they were died at day 21

Answer: The required specification is given in the figure and in the figure caption.

Round 2

Reviewer 1 Report

Dear Authors, 

Thank you for your response. 

In response to my comment "Results Recommended to be paragraphed as findings with proper sub-headings, and not as Exp 1 and 2", you explained Exp1 and 2 (adding a line) without actually diving the results based on findings (not very easy for readers). The results need well structured. 

In response to my comment on soil water content (SWC), I did not find these authors Hans Selye, Fox et al. (Tree Physiology, 2018) cited, as you mentioned in your response. Rehschuh et al. (Plant Physiology, 2020), (Pashkovskiy et al., Environmental and Experimental Botany, 2022) although cited, should also be used as reference in the manuscript (appropriately in results and/or discussion) to explain your answer to my comment. The explanation you have given should be appropriately integrated in the manuscript with all the four citations you mentioned. 

The conclusion looks like a summary of your results. Please improve correlating your findings with earlier research findings available, and your opinion how it might contribute to the scientific community and global climate change. Moreover, please add what appropriate future studies can be done to carry forward your findings. 

In the context of my comment on requesting you to subsection the Discussion section, it would have brought much clarity and easy understanding for the readers even if it is not generally practiced in Biomolecules (as responded from your end). 

Thank you 

Author Response

Dear Authors, Thank you for your response. 

Comment 1. In response to my comment "Results Recommended to be paragraphed as findings with proper sub-headings, and not as Exp 1 and 2", you explained Exp1 and 2 (adding a line) without actually diving the results based on findings (not very easy for readers). The results need well structured. 

Answer 1.  Dear Reviewer, thank you for your evaluation of our work and for your valuable comments. We take into account all your recommendations and improved the manuscript.

According to your recommendations, we divided the Results sections to 3 subsections based on findings, with Section 3.1 and Section 3.2 corresponding to Experiment 1 and Section 3.3 corresponding to Experiment 2.

 Comment 2. In response to my comment on soil water content (SWC), I did not find these authors Hans Selye, Fox et al. (Tree Physiology, 2018) cited, as you mentioned in your response. Rehschuh et al. (Plant Physiology, 2020), (Pashkovskiy et al., Environmental and Experimental Botany, 2022) although cited, should also be used as reference in the manuscript (appropriately in results and/or discussion) to explain your answer to my comment. The explanation you have given should be appropriately integrated in the manuscript with all the four citations you mentioned. 

Answer 2: We added the rationale to use plant-based parameters to characterize the degree of water stress severity in the beginning of Discussion section.

Comment 3. The conclusion looks like a summary of your results. Please improve correlating your findings with earlier research findings available, and your opinion how it might contribute to the scientific community and global climate change. Moreover, please add what appropriate future studies can be done to carry forward your findings.

Answer 3. We added the required information to the Conclusion section.

Comment 4. In the context of my comment on requesting you to subsection the Discussion section, it would have brought much clarity and easy understanding for the readers even if it is not generally practiced in Biomolecules (as responded from your end). 

Answer 4: We divided the Discussion section to 3 subsections.